# High level non-carbapenemase carbapenem resistance by overlaying mutations of *mexR*, *oprD,* and *ftsI* in *Pseudomonas aeruginosa*

Yan Yang,[1,2] Xue Li,[1,2] Lang Sun,[1,2] Xiu-Kun Wang,[1,2] You-Wen Zhang,[1,2] Jing Pang,[1,2] Guo-Qing Li,[1,2,3] Xin-Xin Hu,[1,2] Tong-Ying Nie,[1,2] Xin-Yi Yang,[1,2] Jian-Hua Liu,[4] Gerrit Brandis,[5] Xue-Fu You,[1,2,3] Cong-Ran Li[1,2]

**ABSTRACT**  Carbapenem-resistant *Pseudomonas aeruginosa* (CRPA) is a global threat, but the mechanism of non-carbapenemase carbapenem resistance is still unclear. In the current study, we investigated the contributions of point mutations in *mexR*, *oprD*, and *ftsI* to carbapenem resistance in *P. aeruginosa* during *in vivo* evolution studies with consecutive clinical isolates. Real-time qPCR and Electrophoretic Mobility Shift Assay demonstrated that MexR (Gln55Pro) mutation increased MexAB efflux pump genes expression by altering MexR's binding capacity, leading to a four- to eight-fold increase in meropenem MIC in the Pae d1 Green Δ*mexR* and PAO1Δ*mexR* mutants. The OprD (Trp415*) truncation affected porin structure, and the constructed mutant Pae d1 Green *oprD* Trp415* increased meropenem MIC by 16-fold (from 0.25 to 4 µg/mL). The contribution of *ftsI* mutation to meropenem resistance was confirmed by clinical linkage analysis and was estimated to cause a two-fold increase in meropenem MIC by comparing the resistant clinical isolate with the Pae d1 Green *oprD* Trp415*Δ*mexR* double mutant. The study found that the oprD Trp415* allele alone accounts for the imipenem MIC in clinical isolates, while the Δ*mexR* and *ftsI* Arg504Cys alleles do not contribute to imipenem resistance. In conclusion, we identified and explored the contributions of *mexR*, *oprD,* and *ftsI* mutations to high level non-carbapenemase carbapenem resistance in *P. aeruginosa*. These findings highlight the interplay of different mutations in causing non-carbapenemase carbapenem-resistance in *P. aeruginosa*.

**IMPORTANCE**  The emergence of carbapenem-resistant *Pseudomonas aeruginosa* (CRPA) poses a significant global health threat, complicating treatment options for infections caused by this pathogen. Understanding the mechanisms behind non-carbapenemase carbapenem resistance is critical for developing effective therapeutic strategies. This study provides crucial insights into how specific point mutations in key genes-*mexR*, *oprD*, and *ftsI*-contribute to carbapenem resistance, particularly the MexR (Gln55Pro) mutation's effect on efflux pump expression and the OprD (Trp415*) truncation's impact on porin structure. The findings elucidate the complex interplay of these mutations, highlighting their roles in conferring high-level resistance, and underscore the imperative for continued research to inform therapeutic strategies against CRPA infections.

**KEYWORDS**  *Pseudomonas aeruginosa*, carbapenem resistance, *in vivo* evolution, overlaying point mutation

*P*seudomonas aeruginosa is a Gram-negative opportunistic pathogen and one of the high-priority ESKAPE (*Enterococcus faecium*, *Staphylococcus aureus*, *Klebsiella pneumoniae*, *Acinetobacter baumannii*, *P. aeruginosa*, and *Enterobacter* species) organisms

**Peer Reviewer** Feng Yang, Tenth People's Hospital of Tongji University, Shanghai, China

Address correspondence to Cong-Ran Li, congranli@imb.pumc.edu.cn, Xue-Fu You, xuefuyou@imb.pumc.edu.cn, Gerrit Brandis, gerrit.brandis@icm.uu.se, or Jian-Hua Liu, l_jianhua1979@126.com.

Jian-Hua Liu, Gerrit Brandis, Xue-Fu You, and Cong-Ran Li contributed equally to this article.

The authors declare no conflict of interest.

See the funding table on p. 13.

that are often found to be resistant to multiple antibiotics (1). Carbapenems, a class of β-lactam antibiotics with a broad spectrum of activity against many Gram-positive and Gram-negative bacteria, including those resistant to other β-lactams. Carbapenems enter Gram-negative bacteria via outer membrane proteins (porins) and inhibit penicillin-binding proteins (PBPs), enzymes critical for the final stages of peptidoglycan synthesis in the bacterial cell wall. By inhibiting PBPs, carbapenems disrupt the cross-linking of peptidoglycan chains, weakening the cell wall. This disruption leads to the loss of structural integrity, causing the bacterial cell to burst and die due to osmotic pressure (2). Commonly used members of the carbapenem class in clinical settings include imipenem, meropenem, ertapenem, and doripenem (3). Unfortunately, *P. aeruginosa* can develop resistance to most clinically available carbapenems, which poses a significant threat to public health. Carbapenem-resistant *P. aeruginosa* is classified as "High group" on the WHO priority pathogen list (2024) for R&D of new antibiotics (4). *P. aeruginosa* can become resistant to carbapenems (including imipenem and meropenem) through multiple mechanisms, including overexpression of efflux pumps, reduced outer membrane permeability, production of antibiotic-inactivating enzymes and biofilm-mediated resistance (5, 6), and high-level carbapenem resistance is often mediated by carbapenemases (7). Carbapenemases are specific β-lactamases with the ability to hydrolyze carbapenems and can be classified into three main classes based on their molecular structure: class A carbapenemases (e.g., KPC and GES enzymes), class B carbapenemases (e.g., VIM, IMP, and NDM β-lactamases), and class D carbapenemases (e.g., OXA-type enzymes) (3). In *P. aeruginosa*, metallo-β-lactamases (MBLs), particularly the VIM and IMP types, are the most common carbapenemases (8). However, the mechanism of drug resistance of *P. aeruginosa* is more complex in the absence of carbapenemase enzymes (9, 10). Previous studies have used clinical isolates with differing degrees of carbapenem susceptibility to identify genes that potentially contribute to drug resistance in the absence of efficient carbapenem enzymes. These studies showed that high levels of carbapenem resistance may be the result of the interplay of factors such as *oprD* deletion, increased antibiotic efflux, and elevated expression of *ampC* gene which encodes a cephalosporinase, can degrade cephalosporins, and contribute to broader β-lactam resistance (7, 11, 12). There may also be some poorly understood determinants, such as mutations in the *opdP* gene that encodes a porin (13). However, the full concert of factors that contribute to carbapenem resistance of *P. aeruginosa* in the absence carbapenemase remains to be defined.

In the current study, *in vivo* evolution study of *P. aeruginosa* using consecutive clinical isolates found overlaying point mutations in *mexR*, *oprD,* and *ftsI* of the high level non-carbapenemase carbapenem-resistant clinical *P. aeruginosa*. We validated the specific contributions of the identified mutations to carbapenem resistance in *P. aeruginosa*. These findings will possibly aid in the control and management of carbapenem-resistant *P. aeruginosa* infections.

## RESULTS

### Phenotypes of the clinical *P. aeruginosa* isolates

Pulsed Field Gel Electrophoresis (PFGE) analysis showed identical patterns of the strains isolated at different time points, indicating a common genetic origin (Fig. 1A). Notably, Pae d1 Brown had only one band, whereas the other strains had two bands at sizes of about 78 kb (Fig. 1A). To further investigate this phenomenon, we used SnapGene software to analyze the *SpeI* restriction sites on the genomes of all strains. We found that, except for the Pae d1 Brown strain which has 49 *SpeI* sites on its chromosome, the other strains have 52 *SpeI* sites. Therefore, we hypothesized that the missing *SpeI* site in the Pae d1 brown strain might be associated with the disappearance of the bands. Comparative genomic analysis indicated that all carbapenem-resistant isolates (isolates from day 14 and day 17) and Pae d1 Brown originated from Pae d1 Green, the carbapenem-resistant brown mutants were not from Pae d1 Brown (data not shown), which could explain the disappearance and reappearance of the band.

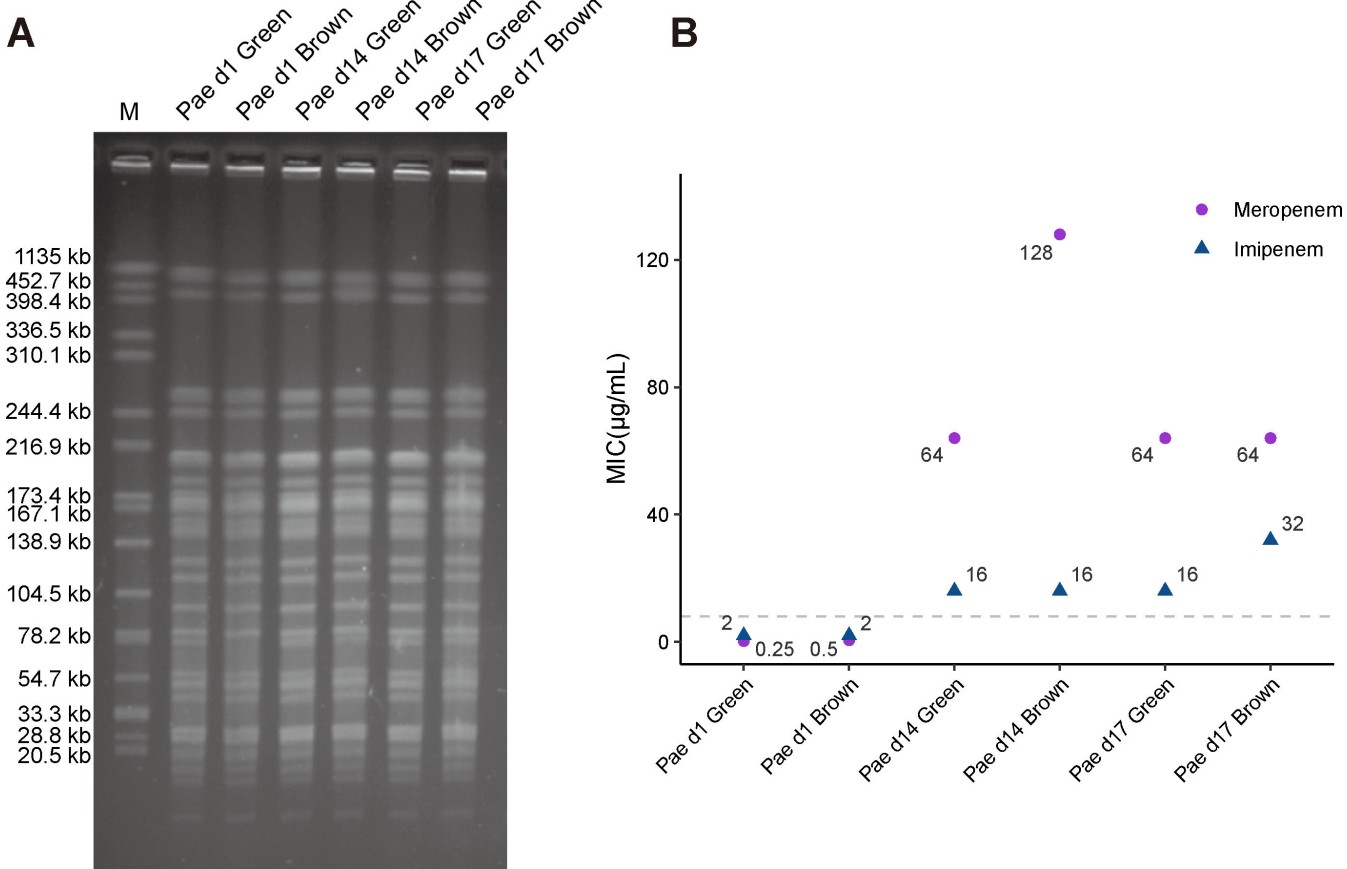

**FIG 1** Phenotypes of *P. aeruginosa* isolates. (A) Pulsed-field gel electrophoresis (PFGE) analysis of the strains. M, (marker) strain *Salmonella enterica* serotype Braenderup H9812 digested with XbaI. (A) The MICs of the isolates as determined by broth microdilution. Dashed line represents the clinical breakpoints from Clinical and Laboratory Standards Institute (CLSI) (M100 30th edition).

The susceptibility of the isolates to carbapenems (meropenem and imipenem) was determined by broth microdilution method. For meropenem, the MICs (minimum inhibitory concentration) of the isolates (Fig. 1B) from day 1 (Pae d1 Green and Pae d1 Brown, 0.25–0.5 µg/mL) were below the clinical breakpoint of 8 µg/mL. However, MICs of isolates after meropenem treatment (Pae d14 Green, Pae d14 Brown, Pae d17 Green, and Pae d17 Brown) were 64 or 128 µg/mL (Fig. 1B). Consistently, the later strains also showed elevated MICs (8–16 times higher than those of the isolates from day 1) to imipenem, another carbapenem antibiotic (Fig. 1B). The resistant isolates from days 14 and 17 were sequentially passaged on LB agar plates in the absence of antibiotic to test for the possible presence of phenotypic adaptation, inducible adaptation, and heteroresistance (14–16). No changes in MICs for meropenem and imipenem were found after seven growth cycles (Table S3), indicating stable carbapenem resistance mutations. Comparative genomic analysis indicated that while all isolates from day 17 shared a common ancestor, they were not direct descendants of the strains that were isolated at day 14 (data not shown), which could explain why the extent of meropenem resistance from day 17 may be different from the isolates from day 14.

## Identification of carbapenem resistance mutation sites

Whole-genome sequencing revealed that all *P. aeruginosa* isolates harbored an about 6.5 Mb-long chromosome belonging to the very rare sequence type ST264 which has not yet been documented in the *Pseudomonas* Genome Database (https://www.pseudo-monas.com). No plasmids were found in the isolates and none of the strains harbored

a carbapenemase gene. Genomic analysis showed that the resistant isolates from days 14 and 17 had identical mutations in four loci: (i) *mexR* Gln55Pro that encodes the multidrug resistance efflux pump repressor MexR, (ii) *oprD* Trp415* encoding the outer membrane porin OprD, (iii) *ftsI* Arg504Cys encoding penicillin-binding protein 3, and (iv) *shaC* Pro279Gln encoding a sodium hydrogen antiporter (Fig. 2A). These four loci have previously been implicated as potentially beneficial mutations for the adaptation of strains to new environments in human hosts according to signatures of parallel evolution (17, 18). Mutations in *mexR*, *oprD*, and *ftsI* have been associated with antibiotic resistance development in *P. aeruginosa* (5). ShaC is part of a Na⁺/H⁺ antiporter that is largely responsible for Na⁺ extrusion in *P. aeruginosa* and has a role in the infection of the pathogen but has not been implicated in relation to carbapenem resistance (19, 20). Thus, genomic information of 228 meropenem susceptible and 270 meropenem resistant strains were obtained from the PATRIC database (Data set S1) to determine if mutations in *shaC* are associated with the development of carbapenem resistance. No linkage between mutations in *shaC* and meropenem resistance was found (Fig. 2B). Thus, *shaC* was excluded from further analysis, and *mexR*, *oprD*, and *ftsI* were chosen for further mechanistic studies on carbapenem resistance in *P. aeruginosa* isolates.

## MexR (Gln55Pro) mutation mediates high expression of *mexA/B* efflux pump genes and reduces susceptibility to meropenem

There are four efflux pump systems in *P. aeruginosa* that have been implicated with reduced β-lactam susceptibility: MexAB-OprM, MexCD-OprJ, MexEF-OprN, and MexXY-OprM (5). MexR, a transcriptional regulator of MarR family, is a negative regulator of the MexAB-OprM efflux pump operon and also negatively regulates its own expression (21). The mutation in the *mexR* gene of the resistant isolates results in a Gln55Pro substitution within the middle of alpha helix α3 of MexR that is an important part of the DNA-binding regulatory domain (22). The Gln55Pro mutation is expected to disrupt formation of helix α3, thus inhibiting MexR activity which is expected to result in increased expression of MexAB-OprM and MexR. To test this hypothesis, real-time qPCR was performed (Fig. S2). As expected, the results showed that the four resistant isolates from day 14 and 17 displayed a 6- to 13- fold increased expressions of the *mexA* and *mexB* genes compared to the susceptible ancestor Pae d1 Green. Expressions of other efflux pump genes (*mexD*, *mexE*, and *mexX*) generally remained unchanged, while the expression of *mexX* was undetectable in all brown isolates since *mexXY* was part of the deleted region (Fig. S2). To

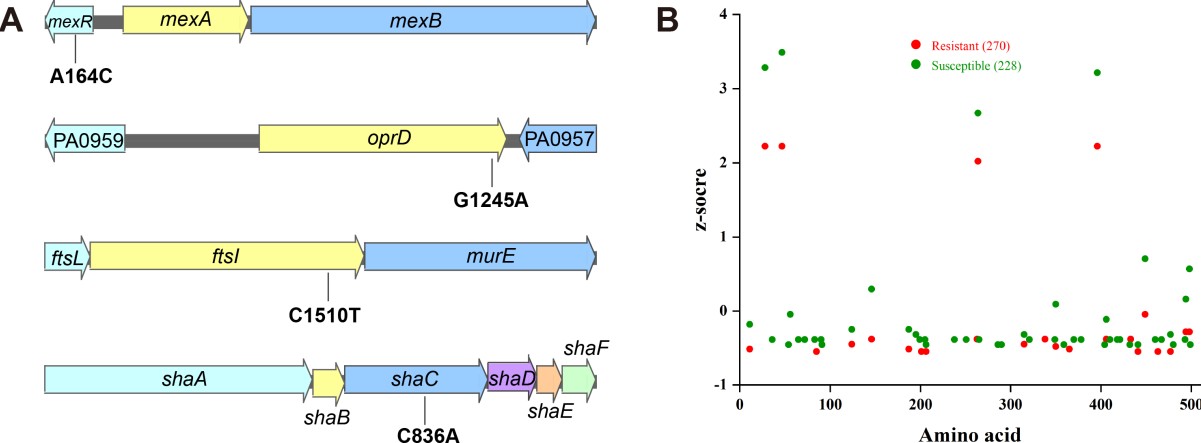

**FIG 2** High level non-carbapenemase carbapenem resistance of *P. aeruginosa* can be possibly explained by the combined effect of *mexR*, *oprD*, and *ftsI* mutations. (A) Shared mutations in carbapenem-resistant bacteria. (B) Mutations in ShaC (Pro279Gln) are not associated with meropenem resistance. ShaC sequences from 270 meropenem resistant (red dots) and 228 susceptible isolates (green dots) were aligned and compared to PAO1, and the total number of variants per residue was calculated for each group of genomes. The total number of amino acid changes was normalized and presented as a *z*-score for each group, plotted against the ShaC residue number.

elucidate the function of the *mexR* Gln55Pro mutation, we constructed a *mexR* deletion mutant of the *P. aeruginosa* model strain PAO1 (PAO1Δ*mexR*) using the pCasPA/pACRISPR system (23). Additionally, plasmids expressing various *mexR* alleles were constructed for a complementation evaluation assay. Deletion of *mexR* resulted in a six fold increase of *mexA* and *mexB* expressions in the PAO1Δ*mexR* strain which was restored back to the wild-type level upon complementation with plasmid-encoded wild-type *mexR* (Fig. 3A). However, complementation with the plasmid-encoded *mexR* Gln55Pro allele did not reduce expression levels of *mexA* or *mexB* (Fig. 3A). Accordingly, MIC measurements demonstrated that deletion of *mexR* led to four fold increase in meropenem MIC which was partially restored upon complementation with wild-type *mexR* but not by the *mexR* Gln55Pro allele (Fig. 3B). These results indicate that the *mexR* Gln55Pro allele results in an inactive MexR protein. An electrophoretic mobility shift assays (EMSA) was conducted to test if this inactivity is due to reduced DNA-binding activity of MexR to the regulatory

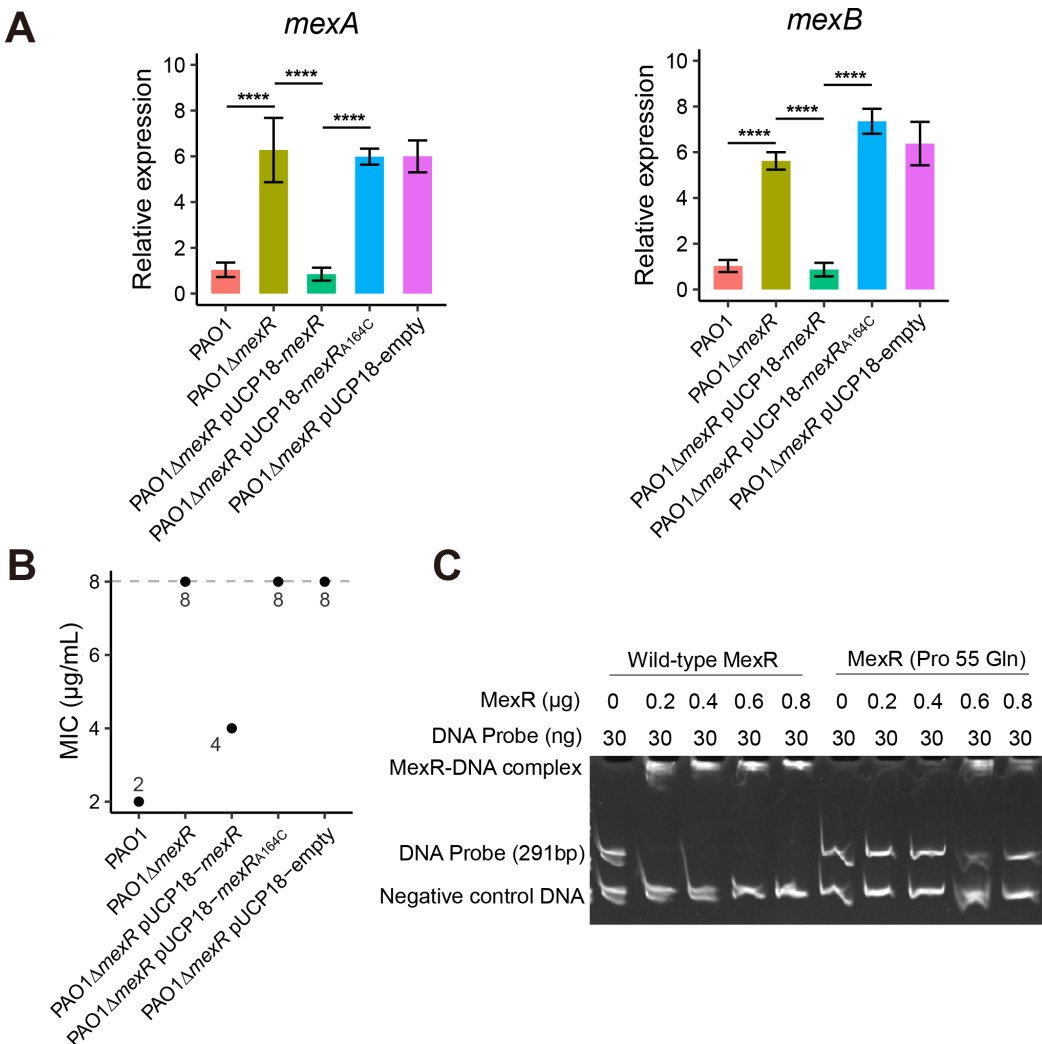

**FIG 3** Mutations in MexR (Gln55Pro) are associated with meropenem resistance. (A) Effect of *mexR* knockout on the expression of *mexA/B* efflux pump genes as determined by real-time qPCR. ****$P < 0.0001$, by two-tailed *t*-test. At least three biological replicates were performed. Bars represent the mean relative expression ± SD. (B) Effect of *mexR* knockout on the susceptibility to meropenem. Dashed line represents the clinical breakpoints from CLSI (M100 30th edition). (C) Electrophoretic mobility shift assay (EMSA) of the interaction between MexR and the target sequence in 20 µL reaction system. Different concentrations of MexR-His (wild-type or mutated) proteins were co-incubated with 30 ng 291 bp intergenic region fragment of *mexR-mexAB* (upstream regulatory region of *mexAB* DNA probe), and 20 ng DNA fragment of the 141 bp partial *rpsL* coding gene was used as a negative control.

region of the *mexA/B* operon (Fig. 3C). Wild-type MexR was able to bind the regulatory region of the *mexA/B* operon, while no binding was observed for MexR Gln55Pro. Finally, the *mexR* gene was deleted in the Pae d1 Green isolate to confirm that inactivation of MexR also results in decreased susceptibility to meropenem in the clinical isolate. As expected, deletion of *mexR* leads to an eight fold increase in the MIC of meropenem in Pae d1 Green (Fig. 4B). These results demonstrate that the clinically relevant MexR Gln55Pro allele leads to high expression of the MexAB-OprM efflux pump of *P. aeruginosa* by abolishing the DNA-binding ability of MexR to its target sequence which results in reduced susceptibility of the strain to carbapenem antibiotics.

## OprD (Trp415*) mutation is associated with meropenem resistance

OprD (OccD1) is a substrate-specific porin in the outer membrane of *P. aeruginosa* (24). According to the X-ray crystal structure analysis, OprD is a monomeric 18-stranded β-barrel (25). The mutation identified in the clinical isolates (*oprD* Trp415*) leads to premature termination resulting in a C-terminal truncation and loss of two strands (β17 and β18) of the β-barrel structure (Fig. 5A). This truncation would be expected to interfere with functional protein assembly in the outer membrane. The OprD protein sequence in Pae d1 Green is not identical to the one found in the laboratory strain PAO1 but shows variations in multiple loci (Fig. 5B). Thus, only the Pae d1 Green strain was used to confirm the role of the *oprD* termination mutation on carbapenem susceptibility. Introducing the *oprD* Trp415* mutation into Pae d1 Green resulted in a 16-fold elevated MIC against meropenem (Fig. 5C). This MIC increase was indistinguishable from the increase resulting from a deletion of the entire *oprD* gene. Complementation with pUCP18 carrying wild-type *oprD* from Pae d1 Green did not restore susceptibility to carbapenems likely due to difficulties in expressing functional OprD from a plasmid system. However, *in situ* complementation on the chromosome successfully restored wild-type meropenem MIC (Fig. S1; Fig. 5C). These data confirm that the *oprD* mutation found in the clinical isolates contributes to carbapenem resistance by inactivating the OprD porin within the cellular membrane.

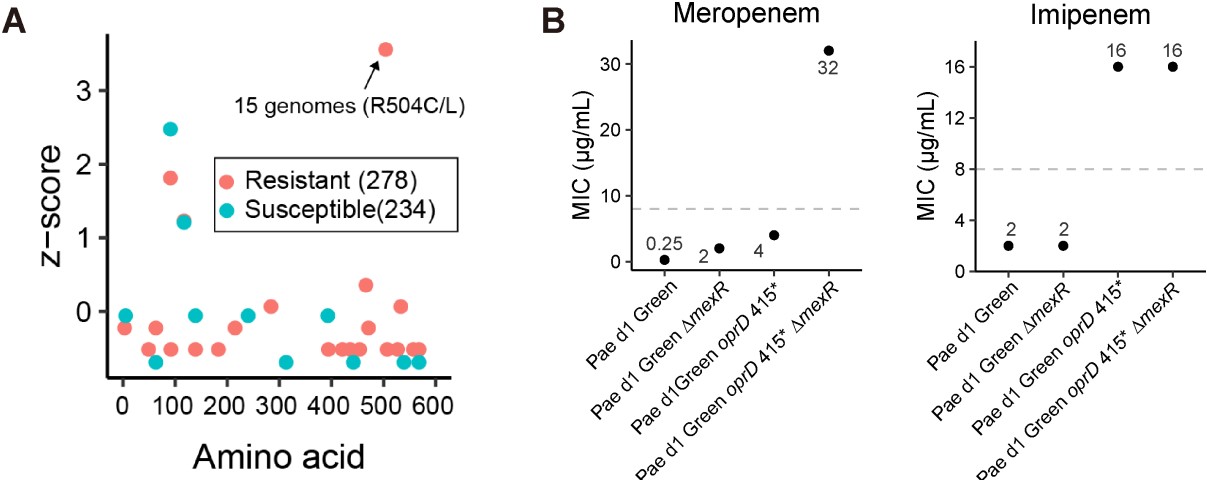

**FIG 4** FtsI (Arg504Cys) mutation is associated with meropenem resistance. (A) The FtsI (Arg504Cys) allele is tightly linked to clinical meropenem resistance. FtsI sequences from resistant (red dots) and susceptible isolates (green dots) were aligned to that from the wild-type Pae d1 Green/Brown by SnapGene® 6.0.2, and the total number of variations per residue was calculated for each genomic group. The total number of amino acid changes was normalized using IBM SPSS Statistics 25 software and presented as *z*-scores for each group. (B) Synergistic impact of *mexR* and *oprD* mutations on meropenem resistance. Dashed line represents the clinical breakpoints from CLSI (M100 30th edition).

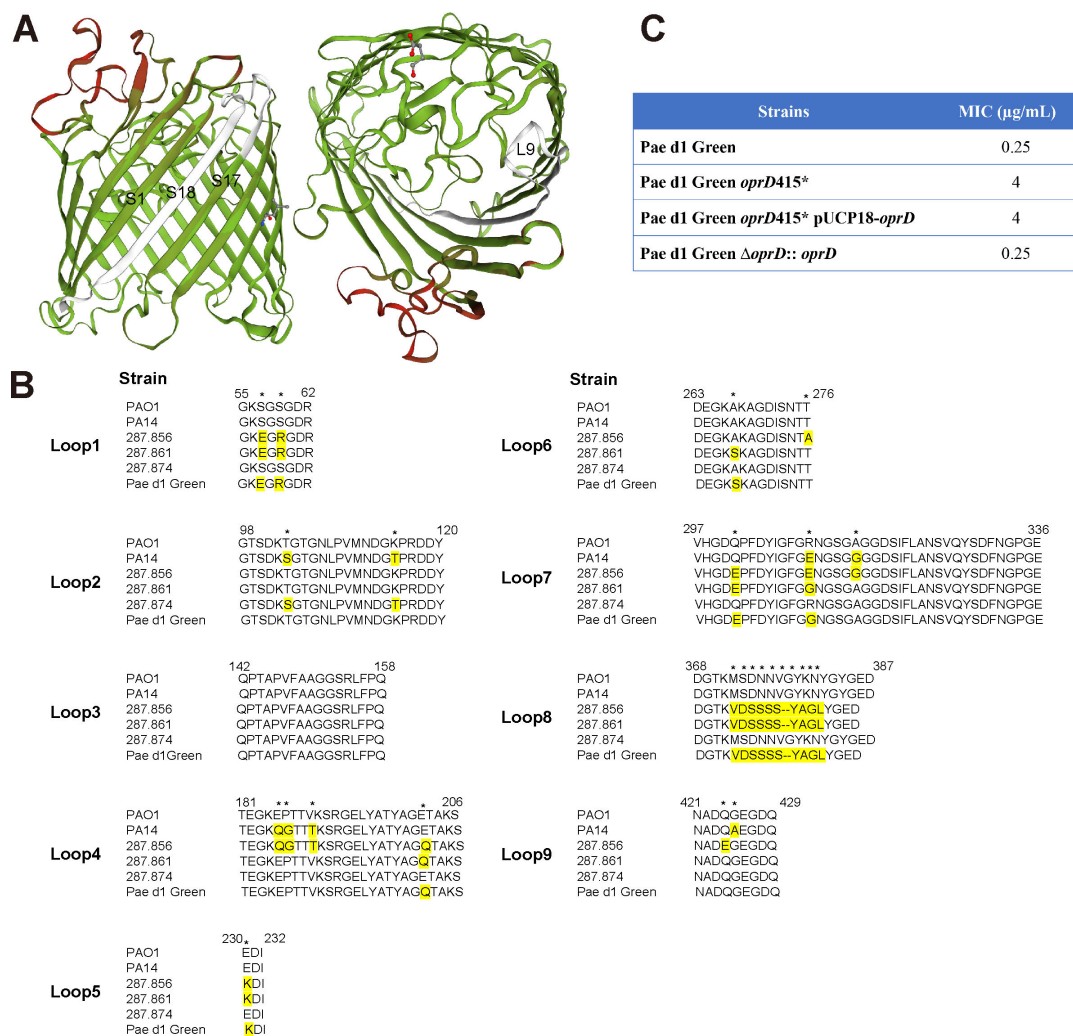

**FIG 5** OprD (Trp415*) mutation is associated with meropenem resistance. (A) Structure of OprD (Occd1) protein from the resistant strains (Pae d14 or Pae d17) compared with that from the susceptible strains (Pae d1 Green/Brown) using SWISS-Model software (https://swissmodel.expasy.org/). Cartoon of OprD viewed from the side (left) and from the extracellular environment (right) are shown. The β-strands S1, S17, and S18, loop L9 are indicated. Mismatches are enhanced by white color. (B) Multiple sequence alignment (Clustal X) of the OprD loops based on the *P. aeruginosa* PAO1 topological model. Amino acids that differ from PAO1 wild-type OprD are highlighted in yellow, the differential amino acid sites between isolates are indicated by stars. Numbering of the amino acids corresponds to the amino acid sequence of the wild-type *P. aeruginosa* PAO1 OprD. PAO1, model strain *P. aeruginosa* PAO1; PA14, model strain *P. aeruginosa* PA14; 287.856, 287.861, 287.874, clinical strains download from PATRIC database (https://www.patricbrc.org/). (C) Effect of *oprD* mutation on the susceptibility to meropenem.

## FtsI (Arg504Cys) mutation may contribute with a twofold increase in the MIC of meropenem

The gene *ftsI* encodes penicillin-binding protein 3 has previously been implicated with carbapenem resistance (26). However, *ftsI* is an essential gene and the construction of the mutant allele in PAO1 or Pae d1 Green was not successful. To determine the significance of the *ftsI* mutation during the evolution of meropenem resistance, we analyzed the genomes of the meropenem resistant and sensitive isolates retrieved from the PATRIC database. Of the meropenem resistant isolates, 15 genomes had an amino acid change at residue number 504 (Arg504Cys/Leu), with a *z*-score of 3.56 (Data set S1 ). In comparison, *ftsI* Arg504Cys/Leu mutations were not found in meropenem susceptible bacteria (Fig. 4A; Data set S1 ). The results indicate that the *ftsI* Arg504Cys mutation found in this study significantly contributes in the development of meropenem resistance. To further test the potential impact of the *ftsI* mutation, a Pae d1 Green strain was constructed

**TABLE 1** Effects of spermine or PAβN on the susceptibility of *P. aeruginosa* to carbapenems

| Strains | MIC (µg/mL) | | | | | |
|---|---|---|---|---|---|---|
| | Meropenem | Meropenem + Sp[a] | Meropenem + PAβN[b] | Imipenem | Imipenem + sp | Imipenem + PAβN |
| Pae d1 Green | 0.25 | 0.5 | 0.25 | 2 | 2 | 0.5 |
| Pae d1 Brown | 0.5 | 0.5 | ≤0.125 | 2 | 1 | ≤0.125 |
| Pae d14 Green | 64 | 8 | 32 | 16 | 4 | 16 |
| Pae d14 Brown | 128 | 16 | 2 | 16 | 2 | 0.25 |
| Pae d17 Green | 64 | 8 | 32 | 16 | 4 | 16 |
| Pae d17 Brown | 64 | 8 | 16 | 32 | 2 | 2 |
| Pae d1 Green Δ*mexR* | 2 | 4 | 1 | 2 | 2 | 0.5 |
| Pae d1 Green *oprD*415* | 4 | 1 | 0.5 | 16 | 2 | 1 |
| Pae d1 Green *oprD*415*Δ*mexR* | 32 | 4 | 2 | 16 | 2 | 1 |
| PAO1 | 2 | 2 | 2 | 4 | 16 | 4 |
| PAO1 Δ*mexR* | 8 | 8 | 8 | 4 | 16 | 4 |
| *P. aeruginosa* ATCC 27853 | 0.5 | 1 | 1 | 2 | 16 | 2 |

[a]Spermine (Sp; 10 mM).
[b]PAβN (20 µg/mL) was used for MIC measurements. The concentrations of spermine or PAβN did not affect the growth of isolates.

that harbored both the Δ*mexR* and the *oprD* Trp415* alleles. The MIC of meropenem was measured for the double mutant and compared to the MIC of the clinical isolates that also harbor the mutant *ftsI* allele. The results show an additive effect for the two genes where the Δ*mexR* allele increases MIC of meropenem 8-fold, the *oprD* Trp415* allele 16-fold, and the MIC of the double mutant is increased 128-fold (Fig. 4B). The MICs of meropenem in the clinical isolates (Pae d14 Green, Pae d17 Green) that also carry the *ftsI* mutation were at least 256-fold higher than that of Pae d1 Green indicating that the *ftsI* Arg504Cys mutation contributes with a twofold increase in the MIC of meropenem (assuming strictly additive contributions). The MICs of imipenem were determined for the constructed strains to test the contribution of the various mutations on decreased susceptibility to another carbapenem antibiotic. The results (Fig. 4B) show that the *oprD* Trp415* allele is necessary and sufficient to explain the imipenem MIC found in the clinical isolates and that the Δ*mexR* and *ftsI* Arg504Cys alleles do not contribute to imipenem resistance.

## Spermine or PAβN effects on MICs of carbapenems in *P. aeruginosa*

We assessed the sensitivity of strains to meropenem and imipenem in the presence or absence of spermine or the efflux pump inhibitor PAβN (Table 1). Spermine significantly enhanced the susceptibility to imipenem and meropenem in clinically carbapenem resistant strains (both brown and green), reducing the MICs of meropenem by 8-fold and imipenem by 4- to 16-fold. Spermine had no effect in the Pae d1 Green Δ*mexR* strain, but it lowered the MIC of both antibiotics (by 4- to 8-fold) in the *oprD*-deficient Pae d1 Green *oprD*415* single mutant and the Pae d1 Green *oprD*415* Δ*mexR* double mutant. Consistent with previous report (27), spermine increased the MIC of imipenem in PAO1, but it did not demonstrate apparent effects in the clinical carbapenem susceptible ones, which may be related with the different *oprD* sequence of the strains as spermine was suggested to cause the increase of imipenem MIC in PAO1 by inhibition of OprD. In the clinical carbapenem-resistant isolates and the constructs containing *oprD* mutations, the addition of spermine enhanced the susceptibility to carbapenems. Polyamines have also been shown to increase *P. aeruginosa* sensitivity to β-lactam antibiotics (28), hence the current results in the strains without *oprD* mutation may suggest the combined effects of spermine on sensitization to carbapenems (28) and inhibition of OprD (27).

PAβN significantly enhanced susceptibility of the brown clinical strains to the antibiotics, with MICs decreased by 4- to 64-fold. In contrast, the green strain exhibited a much weaker response. The difference between green and brown strains may be attributed to the absence of the MexXY operon in the brown strain, PAβN was not reported as an inhibitor of MexXY (29), and the absence of MexXY in the brown isolates

may lead to more susceptibility of the strains to pump inhibition by PAβN. The MICs of carbapenems in the Pae d1 Green Δ*mexR* strain decreased by 2- to 4-fold in the presence of PAβN, while the corresponding changes were 8- to 16-fold in the *oprD*-deficient Pae d1 Green *oprD*415* and Pae d1 Green *oprD*415* Δ*mexR* mutant strains, suggesting the involvement of other functions of PAβN. Actually, PAβN has been reported to increase outer membrane permeability in *P. aeruginosa* (30), and this effect may contribute to its sensitization activity in the absence of functional OprD.

## DISCUSSION

Here, combining current genome sequencing technologies, phenotypic profiling, and functional genetics with the continuous isolation of *P. aeruginosa* from an infected patient enabled the identification of important evolutionary events leading to high-level non-carbapenemase carbapenem resistance. Although the drug resistance mechanism of *P. aeruginosa* is frequently studied by various methods (including retrospective analysis of clinical bacteria, continuous passage *in vitro* or *in vivo*, transposon mutagenesis, and omics methods), most studies on carbapenem resistance of *P. aeruginosa* rely on classification of the isolated genotypes without functional verification of the mutations or determination of the specific contributions of different mutations to high-level carbapenem resistance (31). What need to be mentioned here is that we found both green and brown pigment producing isolates on each of the sample from different time point, the brown pigment producing isolate showed the same carbapenem resistant phenotype and related mutation sites as the corresponding green one. Only the green isolates were used in the further study. Initially, we conducted a *mexR* gene single knockout in both *P. aeruginosa* PAO1 and the susceptible strain Pae d1 Green. The *mexR* knockout increased the meropenem MIC from 2 to 8 µg/mL in PAO1 (Fig. 3B) and from 0.25 to 2 µg/mL in Pae d1 Green (Fig. 4B), without affecting the imipenem MIC. We then reconstructed the OprD Trp415* single mutation in Pae d1 Green, which raised the meropenem MIC to 4 µg/mL and the imipenem MIC to 16 µg/mL (Fig. 4B). Additionally, we deleted the *mexR* gene in the Pae d1 Green OprD415* strain, constructing the double mutant Pae d1 Green OprD415*Δ*mexR*. This double mutant exhibited a meropenem MIC increase to 32 µg/mL and an imipenem MIC increase to 16 µg/mL (Fig. 4B).

Specifically, we were able to show that the mutated transcriptional regulator MexR (Gln55Pro) had decreased binding affinity to its target DNA sequence. This decreased DNA binding led to overexpression of the efflux pump genes *mexA* and *mexB* resulting in decreased meropenem susceptibility. It is most likely that the change from glutamine to proline at position 55 will break the α-helix 3 (residues 54–59) of MexR that is important for its DNA binding affinity (32, 33). The ability of porin OprD to mediate the diffusion of basic amino acids and carbapenems into cells has attracted extensive attention, and OprD porin loss is associated with reduced susceptibility to carbapenems (34, 35). However, at least 15 OprD protein variants were detected in *P. aeruginosa* (34), suggesting that it is not possible to reliably determine the antibiotic resistance phenotype by comparing genotype data of clinical isolates and model bacteria. In the current study, we reconstructed the identified OprD Trp415* mutant allele in the relevant meropenem susceptible isolate Pae d1 Green. We found that the truncated OprD allele was non-functional (indistinguishable from an OprD deletion), reduced susceptibility to meropenem, and resulted in imipenem resistance above the clinical breakpoint (Fig. 4B). Furthermore, the *oprD* mutation seemed to be the only mutation within the isolated strains that affected imipenem resistance. Penicillin-binding proteins (PBPs) are involved in the synthesis and recycling of peptidoglycans (36), which are key structural components of the cell walls of bacteria. PBP3 is not a common target in screens of resistance mechanisms among clinical isolates (37). However, the clinical significance of PBPs polymorphisms in *P. aeruginosa* has been recognized in recent years. PBP3 is considered a target for adaptive mutations in *P. aeruginosa* isolates from cystic fibrosis patients treated with β-lactam, among them, mutation Arg504Cys identified in our study has been found with the highest prevalence (26). Our data suggest that the

FtsI Arg504Cys allele only has a minor contribution to meropenem resistance. The *ftsI* mutation was estimated to increase the MIC to meropenem 2-fold while the mutations in *mexR* and *oprD* increased the MIC 8-fold and 16-fold, respectively. We were able to show that the combination of the *mexR* and *oprD* mutations in Pae d1 Green increase the MIC to 32 µg/mL which is well above the clinical breakpoint of meropenem (8 µg/mL). Thus, the requirement of the *ftsI* mutation is not immediately obvious but could be explained in two ways: (i) the additional mutation gives the triple mutant a selective advantage over the double mutant under sub-MIC conditions (38) or (ii) the *ftsI* mutation occurred as one of the first two resistance mutations during the evolution. A combination of *ftsI* with either mutation in *mexR* or *oprD* would be expected to result in MICs of 4–8 µg/mL which is below or at the clinical breakpoint. Thus, a third mutation could be required to reach clinical resistance if the *ftsI* mutations occurred early. Using *in vivo* microdialysis after a single intravenous dose of 1 g of meropenem, the maximum free interstitial concentration (mean and standard deviation) of meropenem in infected lung tissue was determined to be 11.4 ± 10.9 µg/mL (39). This concentration would be higher than the MIC of meropenem in either double mutant that contains the mutant *ftsI* allele. Thus, an early occurrence of the *ftsI* allele would be a reasonable explanation for the observed triple mutation (*ftsI*, *mexR*, and *oprD*) in the clinical isolates within this study.

Overall, *in vivo* evolution study of *P. aeruginosa* using consecutive clinical isolates found overlaying point mutations in *mexR*, *oprD,* and *ftsI* of the high level non-carbapenemase carbapenem-resistant clinical *P. aeruginosa*. We identified and validated the specific contributions of the identified mutations to carbapenem resistance. These findings highlight the interplay of different mutations in causing non-carbapenemase carbapenem resistance in *P. aeruginosa* and will possibly aid in the control and management of carbapenem-resistant *P. aeruginosa* infections.

## MATERIALS AND METHODS

### Bacterial isolates and clinical data

Clinical *P. aeruginosa* isolates were consecutively collected from a 75-year-old male patient on day 1 before the administration of meropenem, days 14 and 17 after the administration of meropenem at the First Affiliated Hospital of Hebei North University (Hebei, China), from November 25, 2020 to December 14, 2020 (with the day of Respiratory Intensive Care Unit admission as day 1). Two different types of *P. aeruginosa* were isolated at each time point, producing canonical green and non-canonical dark-brown pigments, respectively; the corresponding isolates were named as Pae d1 Green, Pae d1 Brown, Pae d14 Green, Pae d14 Brown, Pae d17 Green, and Pae d17 Brown (Table S1). *P. aeruginosa* isolates were identified by VITEK2 Compact Bacterial Identification and Monitoring System (bioMérieux, France), Antubio Automatic Microbial Mass Spectrometry Detection System (Autof ms1000, China), and 16S rRNA gene analysis (Primers are listed in Table S2). The isolates were cryopreserved and stored in the Chinese Academy of Medical Sciences Collection Center of Pathogenic Microorganisms (CAMS-CCPM). The laboratory strains and their derivatives from this work are listed in Table S1. Unless otherwise stated, *P. aeruginosa* strains were grown at 37°C in Luria-Bertani (LB) medium with shaking at 180 rpm, or alternatively, on LB agar plates (LB supplemented with 15 g/L agarose) when appropriate.

### Pulsed-field gel electrophoresis

PFGE was performed on *P. aeruginosa* isolates as follows: the genomic DNA of each strain was digested with restriction enzyme *SpeI* overnight at 37°C, and *Salmonella* serotype *Braenderup* strain H9812 was digested with restriction enzyme *Xba I* for 3 h at 37°C (as the DNA size marker). Digested products were electrophoresed in CHEF Mapper XA system (Bio-Rad, USA) for 18.5 h at 14°C, 120 degree angle, with switch times of 5 s and 25 s at 6 V/cm.

## MIC determination

The MICs of antibiotics in the presence or absence of spermine or PAβN were determined by broth microdilution method, and the interpretative criteria were from Clinical and Laboratory Standards Institute (CLSI) document M100 30th (40). *P. aeruginosa* ATCC 27853 was used as the quality control.

## Whole-genome sequencing

Genomic DNA was extracted by TIANamp Bacteria DNA Kit (TIANGEN). The harvested DNA was detected by the agarose gel electrophoresis and quantified by Qubit 2.0 Fluorometer (Thermo Scientific, USA). The whole genome of *P. aeruginosa* isolates was sequenced using PacBio Sequel platform and Illumina NovaSeq PE150 at the Beijing Novogene Bioinformatics Technology Co., Ltd (Beijing, China). *De novo* assembly was performed using SMRT Link v5.0.1 software, and GeneMarkS program version 4.17 (http://topaz.gatech.edu/) was used to retrieve the related coding gene. The *Pseudomonas* Genome Database (https://www.pseudomonas.com/) was used for gene function analysis. The sequence types (ST) of the strains were determined using the assembled contigs to query the Multi-Locus Sequence Typing (MLST) v2.0 (https://cge.food.dtu.dk/services/MLST/). Whole-genome profiling, including antimicrobial resistance genes, was performed by searching for the Comprehensive Antibiotic Research Database (CARD). Genomic alignments between the whole genomes of the isolates and the reference genome were performed using the MUMmer (version 3.22) and LASTZ (version 1.02.00) tools, and SNPs (single-nucleotide polymorphisms), Indel (insertion and deletion), or SV (structural variation) were identified.

## Clinical linkage analysis of *ftsI* or *shaC* mutations to meropenem resistance

The analysis of clinical linkage of *ftsI* or *shaC* mutations to meropenem resistance was performed using method previously described (41). Briefly, Genomic information and antibiotic susceptibility profile of 8,671 *P. aeruginosa* clinical isolates were downloaded from the PATRIC database (https://www.bv-brc.org/) in September 2022. After screening, 512 non-repetitive strains with meropenem susceptibility data available were subjected to further analysis. The FtsI or ShaC protein amino acid sequences of the 512 *P. aeruginosa* isolates were downloaded (Supplementary Data set 1) and compared to reference FtsI or ShaC sequence from *P. aeruginosa* PAO1 using SnapGene 6.0.2 for identifying mismatches and gaps. The non-synonymous changes were recorded for each clinical isolate. The non-synonymous changes at each residue were then summarized to reflect the total number of clinical isolates showing those changes. The raw numbers were further transformed into *z*-scores using IBM SPSS Statistics 25 software to determine whether a particular residue was significantly overrepresented by non-synonymous changes linked to meropenem resistant (MEM[R]) or meropenem susceptible (MEM[S]) phenotypes.

## RT-qPCR gene expression analysis

Overnight cultures of strains of interest were diluted 100-fold in fresh LB broth and grown at 37°C in flasks with orbital shaking at 180 rpm to mid-logarithmic phase (OD$_{600}$ of 0.6–0.8). RNA was collected and purified using RNAprep Pure cell/bacteria kit (TIANGEN), followed by cDNA (complementary DNA) synthesis using FastKing RT kit (TIANGEN). The cDNA was then amplified with PowerUp SYBR Green Master Mix (Applied Biosystems) via a StepOnePlus Real-Time PCR system (Applied Biosystems, USA) following the manufacturer's instructions. RT-qPCR primer sequences for genes of interest and the endogenous reference gene *rpsL* are provided in Table S2. Triple technical replicates and at least three biological replicates per strain were tested. Transcript levels of specific targets were evaluated by the comparative $2^{-\Delta\Delta Ct}$ method with *rpsL* RNA as the endogenous control. The results were plotted as mean ± SD and compared by two-tailed *t*-test.

## Mutant construction and complementation

All genomic deletions and substitutions were made in *P. aeruginosa* strains by a pCasPA/pACRISPR-mediating genome-editing methodology resulting in scarless and markerless mutants as previously described (23). Plasmids and primers used are listed in Table S1 and S2. To delete *mexR* gene, plasmid pCasPA (Addgene) was electroporated into *P. aeruginosa* PAO1, and the transformants were selected by 100 µg/mL tetracycline. After induction by L-arabinose for 2 h, the cells containing the pCasPA plasmid were collected and prepared as the electro-competent cells. Then, the pACRISPR assembled with both appropriate 20 nt spacer designed by the CRISPOR website (http://crispor.tefor.net/) and ~500 bp repair arms on both sides for deletion of *mexR* were electroporated into the cells, and the transformants were selected on LB plates supplemented with 100 µg/mL tetracycline and 150 µg/mL carbenicillin. The plasmids were then cured by counter-selection on 5% sucrose plates, and the correct mutant confirmed by PCR and Sanger sequencing was named PAO1Δ*mexR*. For *mexR* complementation, the wild-type and mutant *mexR* genes were PCR amplified from the chromosomal DNAs of Pae d1 Green and Pae d14 Green, respectively, and assembled into the *Escherichia-Pseudomonas* shuttle vector pUCP18. The plasmids pUCP18-*mexR*, pUCP18-*mexR*$_{A164C}$ generated and the empty vector pUCP18-empty (negative control) were then transformed into PAO1Δ*mexR*, and correct colonies were selected on LB plates supplemented with 150 µg/mL carbenicillin and confirmed by Sanger sequencing following PCR amplifications. Similar method was used to construct *mexR* mutant from Pae d1 Green, and the resulted mutant was named as Pae d1 Green Δ*mexR*.

To construct the A1245G mutation in *oprD* gene of Pae d1 Green, we carried out two-step genome-editing procedure (Fig. S1). First, the *oprD* gene in Pae d1 Green was replaced by partial fragment of kanamycin resistance gene amplified from plasmid pET-30a. Second, the *oprD* gene containing point mutation amplified from Pae d14 Green (mutant) or wild-type *oprD* gene amplified from Pae d1 Green (wild-type) was complemented into the cells using pACRISPR assembled with 20 nt spacer located in partial kanamycin resistance gene and appropriate repair arms to generate the *oprD* point mutation mutant and *in situ* complementary strain. The correct colonies were confirmed by PCR and Sanger sequencing, and the resulted mutant was named as Pae d1 Green *oprD*415*. Pae d1 Green *oprD*415* was also used to further knockout *mexR* gene, and the resulted double mutant was named as Pae d1 Green *oprD*415*Δ*mexR*.

## Purification of recombinant MexR and Electrophoretic mobility shift assay

The full lengths of wild-type and mutant *mexR* were amplified from Pae d1 Green (wild-type) and Pae d14 Green (mutant), respectively, and cloned into pET-30a with 6 His-tag fused on the N terminal. The resultant plasmids were transformed into *E. coli* BL21(DE3) and protein expressions were induced with 0.1 mM isopropyl β-D-thiogalactopyranoside (IPTG) at 37°C. Protein purification was performed by Mag-Beads His-Tag Protein Purification (Sangon Biotech, China) following the manufacturer's instruction. Protein purity was confirmed by Sodium Dodecyl Sulfate-Polyacrylamide Gel Electrophoresis (SDS-PAGE) analysis and quantified by Nanodrop 2000 (Thermo Fisher, USA). A 291 bp DNA fragment covering the regulatory region of *mexAB* was PCR amplified with specific oligonucleotide primers (Table S2) and used as the probe in Electrophoretic Mobility Shift Assay (EMSA). As a negative control, a DNA fragment of the 141 bp partial *rpsL* coding gene was amplified with the indicated primers (Table S2). EMSA was performed as described before with minor modifications (42). Briefly, the DNA probe at 30 ng was allowed to interact with different concentrations of wild-type and mutant MexR proteins in a 20 µL reaction system (12.5 mM Tris-HCl, 1 mM EDTA, 50 mM NaCl, 2.5% glycerol, 20 ng negative control DNA, pH 8.0) at 37°C for 40 min. The samples were then loaded onto a 15% One-Step PAGE Gel Fast Preparation Kit (Vazyme E305-01, China) in ice-cold 0.5 × Tris-Borate-EDTA (TBE) buffer which had been pre-run on ice for 1 h and electrophoresed on ice for 1.5 h at 120 V. The gels were stained with ExRed (Zomanbio) in

0.5 × TBE buffer for 30 min, washed withdeionized $H_2O$ for 1 h, and visualized with a Gel Doc XR + imaging system (Bio-Rad, USA).

## ACKNOWLEDGMENTS

This work was supported by the National Natural Science Foundation of China (82330110, 32141003, 82273980), the National Mega-project for Innovative Drugs (grant number 2019ZX09721001), the CAMS Innovation Fund for Medical Sciences (CIFMS) (2021-I2M-1-030, 2021-I2M-1-039), the National Science and Technology Infrastructure of China (Project No. National Pathogen Resource Center-NPRC-32).

Y .Y., X .L., L .S., X-K. W., Y-W. Z., J. P., G-Q. L., X-X .H., T-Y. N., J-H. L., G. B. and C-R. L. contributed to data acquisition and analysis. Y. Y., X-Y. Y., X-F. Y. and C-R. L. contributed to project conception and study design. Y .Y., J-H. L., G. B., X-F. Y. and C-R .L. wrote and revised the manuscript. All authors reviewed the manuscript and approved the submitted version.

## AUTHOR AFFILIATIONS

[1]Beijing Key Laboratory of Antimicrobial Agents, Institute of Medicinal Biotechnology, Chinese Academy of Medical Sciences & Peking Union Medical College, Beijing, China
[2]Division for Medicinal Microorganisms Related Strains, CAMS Collection Center of Pathogenic CAMS Collection Center of Pathogenic, Beijing, China
[3]State Key Laboratory of Bioactive Substances and Functions of Natural Medicines, Institute of Medicinal Biotechnology, Chinese Academy of Medical Sciences & Peking Union Medical College, Beijing, China
[4]Department of Respiratory Medicine, the First Affiliated Hospital of Hebei North University, Zhangjiakou, China
[5]Department of Cell and Molecular Biology (ICM), Uppsala University, Uppsala, Sweden

## AUTHOR ORCIDs

Yan Yang  http://orcid.org/0000-0001-5369-2459
You-Wen Zhang  http://orcid.org/0000-0002-3985-1393
Jing Pang  http://orcid.org/0000-0002-0538-0327
Cong-Ran Li  http://orcid.org/0000-0002-7505-7525

## FUNDING

| Funder | Grant(s) | Author(s) |
| --- | --- | --- |
| MOST \| National Natural Science Foundation of China (NSFC) | 82330110, 32141003 | Xue-Fu You |
| MOST \| National Natural Science Foundation of China (NSFC) | 82273980 | Congran Li |
| the National Mega-project for Innovative Drugs | 2019ZX09721001 | Xue-Fu You |
| the CAMS Innovation Fund for Medical Sciences | 2021-I2M-1-030 | Xiu-Kun Wang |
| the CAMS Innovation Fund for Medical Sciences | 2021-I2M-1-039 | Xin-Yi Yang |
| the National Science and Technology Infrastructure of China | NPRC-32 | Xin-Yi Yang |

## DATA AVAILABILITY

*P. aeruginosa* genome data used in this study (Fig. 2A and 4A) are available in the PATRIC database (https://www.bv-brc.org/) with the sequence IDs listed in Dataset S1. The raw sequence data reported in this paper have been deposited in the Genome Sequence Archive (44) in National Genomics Data Center under BioProject PRJCA021324, China

National Center for Bioinformation/Beijing Institute of Genomics, Chinese Academy of Sciences (GSA: CRA013527) that are publicly accessible at https://ngdc.cncb.ac.cn/gsa.

## ETHICS APPROVAL

This study does not involve the use of identifiable human material or data. The animal husbandry and experiments were performed according to national standards of laboratory animals in China (43).

## ADDITIONAL FILES

The following material is available online.

### Supplemental Material

**Supplemental data (Spectrum01398-24-s0001.xlsx).** Genomic information of 234 meropenem susceptible and 278 meropenem resistant strains were obtained from the PATRIC database.
**Supplemental material (Spectrum01398-24-s0002.docx).** Tables S1 to S3; Fig. S1 and S2.

### Open Peer Review

**PEER REVIEW HISTORY (review-history.pdf).** An accounting of the reviewer comments and feedback.

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
