## [Reviewer comments · Microbiology Spectrum]

Microbiology Spectrum

High level non-carbapenemase carbapenem resistance by overlaying mutations of *mexR*, *oprD* and *ftsI* in *Pseudomonas aeruginosa*

Yan Yang, Xue Li, Lang Sun, Xiukun Wang, Youwen Zhang, Jing Pang, Guoqing Li, Xinxin Hu, Tongying Nie, Xinyi Yang, Jian-Hua Liu, Gerrit Brandis, Xue-Fu You, and Congran Li

Corresponding Author(s): Congran Li, Chinese Academy of Medical Sciences & Peking Union Medical College

Review Timeline:

Submission Date:	June 9, 2024
Editorial Decision:	August 7, 2024
Revision Received:	October 4, 2024
Accepted:	October 17, 2024

Editor: Minsu Kim

Reviewer(s): Disclosure of reviewer identity is with reference to reviewer comments included in decision letter(s). The following individuals involved in review of your submission have agreed to reveal their identity: Feng Yang (Reviewer #2)

Transaction Report:

DOI: <https://doi.org/10.1128/spectrum.01398-24>

Re: Spectrum01398-24 (High level non-carbapenemase carbapenem resistance led by overlaying mutations of mexR, oprD and ftsI in Pseudomonas aeruginosa)

Dear Prof. Congran Li:

Thank you for the privilege of reviewing your work. Your manuscript was reviewed by two referees. I request that you constructively address these concerns in the form of a revised manuscript.

Please see below for details.

Revision Guidelines

Sincerely,
Minsu Kim
Editor
Microbiology Spectrum

Reviewer #1 (Comments for the Author):

Overall, tremendous effort with very interesting results which I believe will be very much of interest to the readership. Methods & results as presented are concordant. Consider the following as opportunities to provide more insight regarding the importance of these findings

1. Please elaborate on the order of acquisition of various pumps & porins relative to phenotypic profile.
2. Appreciate the effort to address functional aspects of various resistance mechanisms. That said, these assays are fraught with inter- and intra-run / laboratory variability. Please consider extending the studies and confirming functional impact with addition chemical inhibitors of various pumps & porins on the constructs available.

Reviewer #2 (Comments for the Author):

In this study, the authors characterized the mechanism of carbapenemase-independent carbapenem resistance in consecutive clinical isolates treated with meropenem. No plasmid or carbapenemase gene was found. The resistant isolates harbored four common mutations. The authors used several approaches, both experimental (wet) and computational (dry), to validate the contribution of these mutations to resistance. The study is well-designed, and the data is robust. I have only a few minor comments

The abstract contains many lengthy sentences. Please rephrase them

The introduction is overly simplified and requires more background information. For instance, what is the mode of action of carbapenems? Which members of this class are commonly used in clinical settings? How do carbapenemases confer resistance to carbapenems? Additionally, how does the elevated expression of the cephalosporinase-encoding ampC gene confer resistance?

Line 57-58. A reference to a WHO document is needed

Line 80-81. In Figure 1A, one strain (Pae d1 Brown) showed a difference in the bands. It had only one band, whereas the other strains had two bands. There is no noticeable difference in the brightness of the other bands in this strain compared to the others, suggesting that it likely lost the band. However, in later isolates, the band reappeared. How can the disappearance and reappearance of the band be interpreted?

Line 90-91. Although resistance was not lost during in vitro passage in the absence of drug, Fig 1B indicates the extent of meropenem resistance on day 14 is lower than day 7. Authors need to add this data in the text and add discussions on this result.

Line 100-102. Use an arrow to indicate the 6.5 Mb chromosome in Fig 1A.

Line 319. The CLSI document needs a reference.

POINT BY POINT RESPONSE

Manuscript title: High level non-carbapenemase carbapenem resistance by overlaying mutations of *mexR*, *oprD* and *ftsI* in *Pseudomonas aeruginosa*

Manuscript ID: Spectrum 01398-24

Reviewer #1 (Comments for the Author):

Overall, tremendous effort with very interesting results which I believe will be very much of interest to the readership. Methods & results as presented are concordant. Consider the following as opportunities to provide more insight regarding the importance of these findings.

1. Please elaborate on the order of acquisition of various pumps & porins relative to phenotypic profile.

Response: Thank you for your valuable suggestion. We have now expanded the discussion to include a detailed explanation of the sequential acquisition of the various efflux pumps and porin mutations in relation to the phenotypic changes observed. The additions are as follows (Line 265-273): “Initially, we conducted a *mexR* gene single knockout in both *P. aeruginosa* PAO1 and the susceptible strain Pae d1 Green. The *mexR* knockout increased the meropenem MIC from 2 to 8 µg/mL in PAO1 (Fig. 3B) and from 0.25 to 2 µg/mL in Pae d1 Green (Fig. 5B), without affecting the imipenem MIC. We then reconstructed the *oprD* Trp415* single mutation in Pae d1 Green, which raised the meropenem MIC to 4 µg/mL and the imipenem MIC to 16 µg/mL (Fig. 5B). Additionally, we deleted the *mexR* gene in the Pae d1 Green *oprD*415* strain, constructing the double mutant Pae d1 Green *oprD*415* Δ *mexR*. This double mutant exhibited a meropenem MIC increase to 32 µg/mL and an imipenem MIC increase to 16 µg/mL (Fig. 5B).”

2. Appreciate the effort to address functional aspects of various resistance mechanisms. That said, these assays are fraught with inter- and intra-run / laboratory variability. Please consider extending the studies and confirming functional impact with addition chemical inhibitors of various pumps

& porins on the constructs available.

Response: Thank you for your valuable suggestion. We have tested the susceptibility of strains to meropenem and imipenem in the presence or absence of spermine or efflux pump inhibitor PA β N. The results are as follows, and the corresponding information has been added in the revised manuscript (Line 225-251).

Table 1. Effects of spermine or PA β N on the susceptibility of *P. aeruginosa* to carbapenems.

Strains	MIC (μ g/mL)					
	Meropenem	Meropenem + Sp ^a	Meropenem + PA β N ^b	Imipenem	Imipenem+ Sp	Imipenem+ PA β N
Pae d1 Green	0.25	0.5	0.25	2	2	0.5
Pae d1 Brown	0.5	0.5	≤ 0.125	2	1	≤ 0.125
Pae d14 Green	64	8	32	16	4	16
Pae d14 Brown	128	16	2	16	2	0.25
Pae d17 Green	64	8	32	16	4	16
Pae d17 Brown	64	8	16	32	2	2
Pae d1 Green $\Delta mexR$	2	4	1	2	2	0.5
Pae d1 Green oprD415*	4	1	0.5	16	2	1
Pae d1 Green oprD415* $\Delta mexR$	32	4	2	16	2	1
PAO1	2	2	2	4	16	4
PAO1 $\Delta mexR$	8	8	8	4	16	4
P. aeruginosa ATCC 27853	0.5	1	1	2	16	2

^aSpermine (Sp; 10 mM) and ^bPA β N (20 μ g/mL) were used for MIC measurements. The concentrations of spermine or PA β N did not affect the growth of isolates.

We assessed the sensitivity of strains to meropenem and imipenem in the presence or absence of spermine or the efflux pump inhibitor PA β N (Table 1). Spermine significantly enhanced the susceptibility to imipenem and meropenem in clinically carbapenem resistant strains (both brown and green), reducing the MICs of meropenem by 8-fold and imipenem by 4- to 16-fold. Spermine had no effect in the Pae d1 Green $\Delta mexR$ strain, but it lowered the MIC of both antibiotics (by 4- to 8-fold) in the *oprD*-deficient Pae d1 Green *oprD415** single mutant and the Pae d1 Green *oprD415** $\Delta mexR$ double mutant. Consistent with previous report [27], spermine increased the MIC of imipenem in PAO1, but it didn't demonstrate apparent effects in the clinical carbapenem susceptible ones, which may be related with the different *oprD* sequence of the strains as spermine was suggested to cause the increase of imipenem MIC in PAO1 by inhibition of OprD. In the

clinical carbapenem resistant isolates and the constructs containing *oprD* mutations, the addition of spermine enhanced the susceptibility to carbapenems. Polyamines have also been shown to increase *P. aeruginosa* sensitivity to β -lactam antibiotics [28], hence the current results in the strains without *oprD* mutation may suggest the combined effects of spermine on sensitization to carbapenems [28] and inhibition of OprD [27].

PA β N significantly enhanced susceptibility of the brown clinical strains to the antibiotics, with MICs decreased by 4- to 64-fold. In contrast, the green strain exhibited a much weaker response. The difference between green and brown strains may be attributed to the absence of the MexXY operon in the brown strain, PA β N was not reported as an inhibitor of MexXY [29], the absence of MexXY in the brown isolates may lead to more susceptibility of the strains to pump inhibition by PA β N. The MICs of carbapenems in the Pae d1 Green $\Delta mexR$ strain decreased by 2- to 4-fold in the presence of PA β N, while the corresponding changes were 8 to 16-fold in the *oprD*-deficient Pae d1 Green *oprD415** and Pae d1 Green *oprD415* $\Delta mexR$* mutant strains, suggesting the involvement of other functions of PA β N. Actually, PA β N has been reported to increase outer membrane permeability in *P. aeruginosa* [30], and this effect may contribute to its sensitization activity in the absence of functional OprD.

References

- [27] Kwon D-H, Lu C-D. Polyamine effects on antibiotic susceptibility in bacteria. *Antimicrob Agents Chemother* 2007;51:2070–7. <https://doi.org/10.1128/AAC.01472-06>.
- [28] Kwon DH, Lu C-D. Polyamines increase antibiotic susceptibility in *Pseudomonas aeruginosa*. *Antimicrob Agents Chemother* 2006;50:1623–7. <https://doi.org/10.1128/AAC.50.5.1623-1627.2006>.
- [29] Compagne N, Vieira Da Cruz A, Müller RT, Hartkoorn RC, Flipo M, Pos KM. Update on the discovery of efflux pump inhibitors against critical priority Gram-negative bacteria. *Antibiotics (Basel)* 2023;12. <https://doi.org/10.3390/antibiotics12010180>.
- [30] Lamers RP, Cavallari JF, Burrows LL. The efflux inhibitor phenylalanine-arginine beta-naphthylamide (PA β N) permeabilizes the outer membrane of Gram-negative bacteria. *PLoS ONE* 2013;8:e60666. <https://doi.org/10.1371/journal.pone.0060666>.

Reviewer #2 (Comments for the Author):

In this study, the authors characterized the mechanism of carbapenemase-independent carbapenem resistance in consecutive clinical isolates treated with meropenem. No plasmid or carbapenemase gene was found. The resistant isolates harbored four common mutations. The authors used several approaches, both experimental (wet) and computational (dry), to validate the contribution of these mutations to resistance. The study is well-designed, and the data is robust. I have only a few minor comments

The abstract contains many lengthy sentences. Please rephrase them.

Response: We have simplified and restructured the sentences in the abstract to enhance clarity.

The revised abstract is as follows:

“Carbapenem-resistant *Pseudomonas aeruginosa* (CRPA) is a global threat, but the mechanism of non-carbapenemase carbapenem resistance is still unclear. In the current study, we investigated the contributions of point mutations in *mexR*, *oprD*, and *ftsI* to carbapenem resistance in *P. aeruginosa* during *in vivo* evolution studies with consecutive clinical isolates. Real-time qPCR and Electrophoretic Mobility Shift Assay demonstrated that MexR (Gln55Pro) mutation increased *mexAB* efflux pump genes expression by altering MexR's binding capacity, leading to a 4- to 8-fold increase in meropenem MIC in the Pae d1 Green $\Delta mexR$ and PAO1 $\Delta mexR$ mutants. The OprD (Trp415*) truncation affected porin structure, and the constructed mutant Pae d1 Green *oprD* Trp415* increased meropenem MIC by 16 fold (from 0.25 to 4 $\mu\text{g/mL}$). The contribution of *ftsI* mutation to meropenem resistance was confirmed by clinical linkage analysis and was estimated to cause a 2-fold increase in meropenem MIC by comparing the resistant clinical isolate with the Pae d1 Green *oprD* Trp415* $\Delta mexR$ double mutant. The study found that the *oprD* Trp415* allele alone accounts for the imipenem MIC in clinical isolates, while the $\Delta mexR$ and *ftsI* Arg504Cys alleles do not contribute to imipenem resistance. In conclusion, we identified and explored the contributions of *mexR*, *oprD* and *ftsI* mutations to high level non-carbapenemase carbapenem resistance in *P. aeruginosa*. These findings highlight the interplay of different mutations in causing non-carbapenemase carbapenem-resistance in *P. aeruginosa*.”

The introduction is overly simplified and requires more background information. For instance, what is the mode of action of carbapenems? Which members of this class are commonly used in clinical settings? How do carbapenemases confer resistance to carbapenems? Additionally, how does the elevated expression of the cephalosporinase-encoding *ampC* gene confer resistance?

Response: Thank you for your valuable feedback. In the revised manuscript (Line 63-72, 78-82), we have expanded the introduction to include the mode of action of carbapenems, commonly used carbapenems in clinical settings and carbapenemase-mediated resistance, and cited more references. We also added a description (Line 88-89) of how the elevated expression of the *ampC* gene confer resistance, “which encodes a cephalosporinase, can degrade cephalosporins and contribute to broader β -lactam resistance.”

Line 57-58. A reference to a WHO document is needed

Response: Thank you for your suggestion. In the revised manuscript (Line 74), we have added a reference to the relevant WHO document, we also have revised the introduction based on the latest 2024 updated version. In the 2024 version, carbapenem-resistant *P. aeruginosa* is classified as a “High group” on the WHO priority pathogen list.

Line 80-81. In Figure 1A, one strain (Pae d1 Brown) showed a difference in the bands. It had only one band, whereas the other strains had two bands. There is no noticeable difference in the brightness of the other bands in this strain compared to the others, suggesting that it likely lost the band. However, in later isolates, the band reappeared. How can the disappearance and reappearance of the band be interpreted?

Response: We appreciate the reviewer's observation. In the revised manuscript (Line 102-111), we have added the following content: “Notably, Pae d1 Brown had only one band, whereas the other strains had two bands at sizes of about 78 kb (Fig. 1A). To further investigate this phenomenon, we used SnapGene software to analyze the *SpeI* restriction sites on the genomes of all strains. We found that, except for the Pae d1 Brown strain which has 49 *SpeI* sites on its chromosome, the other strains have 52 *SpeI* sites. Therefore, we hypothesized that the missing *SpeI* site in the Pae

d1 brown strain might be associated with the disappearance of the bands. Comparative genomic analysis indicated that all carbapenem resistant isolates (isolates from day 14 and day 17) and Pae d1 Brown originated from Pae d1 Green, the carbapenem resistant brown mutants were not from Pae d1 Brown (data not shown), which could explain the disappearance and reappearance of the band.” These results will be reported later in detail in the *in vivo* evolution study of these isolates.

Line 90-91. Although resistance was not lost during *in vitro* passage in the absence of drug, Fig 1B indicates the extent of meropenem resistance on day 14 is lower than day 7. Authors need to add this data in the text and add discussions on this result.

Response: Thank you for pointing this out. In the revised manuscript, we have added the results of the susceptibility of resistant bacteria to the tested drugs after 7 days of passaging on drug-free plates to Supplementary Material Table S3. And also “Comparative genomic analysis indicated that while all isolates from day 17 shared a common ancestor, they were not direct descendants of the strains that were isolated at day 14 (data not shown), which could explain why the extent of meropenem resistance from day 17 may be different from the isolates from day 14.” The information has been added in the revised manuscript (Line 122-125), and the relating data will be reported later in detail in the *in vivo* evolutionary study of these isolates.

Line 100-102. Use an arrow to indicate the 6.5 Mb chromosome in Fig 1A.

Response: Thank you for your suggestion. We have updated Fig 1A to include the sizes of the marker bands in Fig. 1A. This addition highlights the specific fragment size as requested. In the Pulsed-field gel electrophoresis (PFGE) experiment, we used the *XbaI*-digested *Salmonella enterica* serotype Braenderup H9812 strain as the marker. According to report (<https://doi.org/10.1128/jcm.43.3.1045-1050.2005>), the band sizes of strain H9812 range from 20.5 to 1,135 kb, so we are uncertain about the position of the 6.5 Mb chromosome in Fig. 1A.

Line 319. The CLSI document needs a reference.

Response: We have added the appropriate reference to the CLSI document in the revised manuscript (Line 349).

Re: Spectrum01398-24R1 (High level non-carbapenemase carbapenem resistance by overlaying mutations of *mexR*, *oprD* and *ftsI* in *Pseudomonas aeruginosa*)

Dear Prof. Congran Li:

I am happy to inform you that your manuscript has been accepted, and I am forwarding it to the ASM production staff for publication. Your paper will first be checked to make sure all elements meet the technical requirements. ASM staff will contact you if anything needs to be revised before copyediting and production can begin. Otherwise, you will be notified when your proofs are ready to be viewed.

Sincerely,
Minsu Kim
Editor
Microbiology Spectrum